

# Revisiting the Zingiberales: using multiplexed exon capture to resolve ancient and recent phylogenetic splits in a charismatic plant lineage

Chodon Sass[1], William J.D. Iles[1], Craig F. Barrett[2,3], Selena Y. Smith[4] and Chelsea D. Specht[1]

[1] Department of Plant and Microbial Biology, Department of Integrative Biology and the University and Jepson Herbaria, University of California, Berkeley, Berkeley, CA, United States
[2] Division of Plant and Soil Sciences, West Virginia University, Morgantown, WV, United States
[3] Department of Biology, California State University, Los Angeles, Los Angeles, CA, United States
[4] Department of Earth & Environmental Sciences and the Museum of Paleontology, University of Michigan, Ann Arbor, MI, United States

Corresponding author
Chodon Sass, chodon@berkeley.edu

## ABSTRACT

The Zingiberales are an iconic order of monocotyledonous plants comprising eight families with distinctive and diverse floral morphologies and representing an important ecological element of tropical and subtropical forests. While the eight families are demonstrated to be monophyletic, phylogenetic relationships among these families remain unresolved. Neither combined morphological and molecular studies nor recent attempts to resolve family relationships using sequence data from whole plastomes has resulted in a well-supported, family-level phylogenetic hypothesis of relationships. Here we approach this challenge by leveraging the complete genome of one member of the order, *Musa acuminata*, together with transcriptome information from each of the other seven families to design a set of nuclear loci that can be enriched from highly divergent taxa with a single array-based capture of indexed genomic DNA. A total of 494 exons from 418 nuclear genes were captured for 53 ingroup taxa. The entire plastid genome was also captured for the same 53 taxa. Of the total genes captured, 308 nuclear and 68 plastid genes were used for phylogenetic estimation. The concatenated plastid and nuclear dataset supports the position of Musaceae as sister to the remaining seven families. Moreover, the combined dataset recovers known intra- and inter-family phylogenetic relationships with generally high bootstrap support. This is a flexible and cost effective method that gives the broader plant biology community a tool for generating phylogenomic scale sequence data in non-model systems at varying evolutionary depths.

## INTRODUCTION

Zingiberales are a diverse group of tropical monocots, including important tropical crop plants (e.g., ginger, turmeric, cardamom, bananas) and ornamentals (e.g., cannas, bird-of-paradise, prayer plants). Eight families are recognized with a total of ca. 2500 species.

Fossil zingibers are known since the Cretaceous, and show a mix of characters from Musaceae and Zingiberaceae (*Friis, 1988*; *Rodriguez-de la Rosa & Cevallos-Ferriz, 1994*; *Iles et al., 2015*) on the basis of fruits, seeds, leaves, rhizomes, and phytoliths (*Friis, Crane & Pedersen, 2011*; *Chen & Smith, 2013*). Zingiberales are thought to have diverged from the sister order Commelinales (sensu *Angiosperm Phylogeny Group, 2003*) between 80–124 Ma, with diversification into the major lineages occurring from ca. 60–100 Ma (*Kress & Specht, 2006*; *Magallón et al., 2015*). However, relationships among the families are not well-resolved using multi-gene phylogenies (*Kress et al., 2001*; *Barrett et al., 2014*), likely due to this early rapid radiation. Specifically, the relationship between Musaceae, Strelitziaceae + Lowiaceae, Heliconiaceae, and the remaining four families, which form a well-supported monophyletic group (i.e., the 'ginger clade'), have conflicting support among studies. Whole plastid data for 14 taxa spanning the eight families still failed to resolve the early diverging branches of the phylogeny, perhaps owing to limited sampling and a lack of phylogenetic signal in the plastome (*Barrett et al., 2014*). However challenging to resolve, rapid evolutionary radiations are thought to be a common theme across the tree of life and are thought to explain poorly resolved phylogenies in many groups including insects, birds, bees, turtles, mammals, and angiosperms (*Whitfield & Lockhart, 2007*; *Whitfield & Kjer, 2008*).

The advent of high throughput sequencing and methods that extend the utility of new sequencing technology to non-model organisms has enabled sequence-based understanding of evolutionary relationships in previously intractable groups (*Crawford et al., 2012*; *Faircloth et al., 2012*; *Lemmon, Emme & Lemmon, 2012*; *Bi et al., 2013*). Specifically, for phylogenetic studies, multiple genes containing appropriate levels of sequence divergence can now be obtained for many phylogenetically distant individuals. Various genome enrichment methods, using hybridization to capture a targeted set of genes based on appropriately designed nucleotide probes, have enabled targeted sets of hundreds or thousands of loci to be sequenced in parallel for multiple individuals. However, the ability to capture loci across relatively deep phylogenetic scales has remained challenging because of the inverse relationship between capture efficiency and the evolutionary distance from the individual(s) used to design the probes (*Bi et al., 2012*; *Lemmon, Emme & Lemmon, 2012*; *Peñalba et al., 2014*; *Weitemier et al., 2014*). For very deep divergences in animals, to understand amniote evolution or deep divergences in vertebrate evolution for example, ultra-conserved elements (*Faircloth et al., 2012*) and anchored hybrid enrichment (*Lemmon, Emme & Lemmon, 2012*) have been used to target conserved loci that are flanked by less conserved regions. However, these regions were developed using animal genomes and are unsuitable for use in plants (*Reneker et al., 2012*).

Historical whole genome duplication followed by fractionation and diploidization, genome-level processes that are common during plant evolution and occur in a lineage-specific manner, make it likely that loci with known orthology will need to be tested and developed separately for each plant lineage. Some methods have been developed for lineage specific capture, such as whole exome capture (*Bi et al., 2012*) that

uses a transcriptome sequence and a relatively closely related sequenced genome to design lineage-specific baits. This approach was modified and recently used in plants (*Weitemier et al., 2014*). However, the success of these approaches to capture targeted genes is limited by the distance of the samples to the target transcriptome. A more flexible approach uses PCR products to generate a home-made, in-solution capture (*Maricic, Whitten & Pääbo, 2010*; *Peñalba et al., 2014*), but this requires some prior knowledge of locus sequence and primer optimization and likely is most useful to target 10–50 loci with known phylogenetic utility.

In the case of the Zingiberales, with possibly over 100 Myr of divergence since the initial lineage diversification leading to the modern families, it is necessary to design a set of probes that can capture sequences with a relatively high percentage of polymorphisms, yet still allow the reliable assignment of orthology to captured sequences. In order to do this, we used transcriptomes that were generated as part of the Monocot Tree of Life Project (MonAToL: http://www.botany.wisc.edu/monatol/) or One Thousand Plant Transcriptomes (OneKP: https://sites.google.com/a/ualberta.ca/onekp/home) together with the annotated whole genome of *Musa acuminata* (*D'Hont et al., 2012*) to design a set of probes that were printed on an Agilent microarray chip in parallel. This parallel printing approach enables divergent taxa to be captured on a single array and alleviates binding competition between closely related and divergent individuals. Simultaneously, we captured whole plastid genomes based on published plastid genomes from one member each of the eight families (*Barrett et al., 2014*).

We show the utility of this cost effective method in generating phylogenetically informative sequence data by constructing a phylogenetic tree of the Zingiberales that recaptures known relationships and resolves previously recalcitrant parts of the phylogeny with high support. Because of the phylogenetic breadth of transcriptomes becoming publically available across the plant kingdom, this method has the potential to aid in the design of lineage specific sequencing projects that span phylogenetic distances on the order of 100 Myr or possibly greater.

## METHODS

### Taxon sampling, DNA extraction, and library preparation

Sampling included several members of each of the eight families: Heliconiaceae (5), Musaceae (9, including 2 previously published whole genomes, *D'Hont et al., 2012*; *Davey et al., 2013*), Strelitziaceae (3), Lowiaceae (2), Zingiberaceae (16), Costaceae (10), Marantaceae (7), and Cannaceae (3). In total, 53 individuals were sequenced de novo (Table S1). DNA was extracted using an SDS and salt extraction protocol (*Edwards, Johnstone & Thompson, 1991*; *Konieczny & Ausubel, 1993*) from freshly collected leaves dried in silica, eluted in TE buffer, and sonicated with a Bioruptor® (Diagenode, Liège, Belgium) or qSonica Q800R machine to an average size of approximately 250bp. Sonicated DNA was cleaned and concentrated with solid phase reversible immobilization magnetic beads (Sera-Mag; GE Healthcare, Little Chalfont, UK), and libraries were prepared according to *Meyer & Kircher (2010)*.

## Probe design, sequence capture, sequencing

To generate a nuclear probe set, the *Musa acuminata* CDS was downloaded from the banana genome hub (http://banana-genome.cirad.fr/) and split into annotated exons. Raw reads of transcriptomes for each of the remaining seven families were cleaned to remove adapters, low-complexity sequences, contamination, and PCR duplicates (*Singhal, 2013*). Cleaned transcriptome reads were aligned to the *Musa acuminata* exons using NovoAlign v3.01 (http://novocraft.com) with –t 502 to allow highly divergent sequences to map. After mapping, SNPs were called using SAMtools v0.1.18 (*Li et al., 2009*) and VarScan v2.3.6 (*Koboldt et al., 2012*) and consensus sequences for each family were made based on SNP calls. All exons were filtered for: (1) having overlapping read coverage in all 7 families (2) being longer than 150 bp (3) having between 30–70% GC content (4) being unique by reciprocal BLAST (5) not being found in the RepeatMasker database (command parameters can be found in Supplementary Methods). Although conservation to the *Musa* sequence was not used as a direct filter, the alignment protocol inherently limits chosen regions to those with relative conservation across the families–the minimum percent identity between any *Musa* exon and a family specific bait for the same region was 86% (Table S1). After filtering, a total of 494 exons from 418 genes for each of the eight families (the *Musa* reference sequence plus each sequence from the seven families) were printed with 1 bp tiling twice each on an Agilent 1M microarray chip (G3358A) (Fig. 1A). A second chip was printed with one complete plastid genome from each family (*Barrett et al., 2014*) with slightly less than 1 bp tiling. Libraries from a total of 56 individuals were quantified by Qubit® and pooled in equimolar quantities. The total library pool was split in half and one half was hybridized to the nuclear array and the other half was hybridized to the plastid array (*Hodges et al., 2009*). After hybridization, pools were subject to a limited amount of PCR amplification and enrichment success was verified with qPCR using primers matching both targeted and non-targeted regions. Because of known bias toward plastid dominance in sequenced reads owing to a greater percentage of plastid DNA in the total genomic DNA extractions, the separate hybridization pools were combined in a ratio of 3 parts nuclear to 1 part plastid and sequenced (100 bp paired-end reads) in one lane of a Illumina® HiSeq® 2500 platform at the Vincent J. Coates Genomics Sequencing Facility at the University of California, Berkeley.

## Read processing

Raw reads were cleaned to remove adapters, low-complexity sequences, contamination, and PCR duplicates (*Singhal, 2013*). Custom Perl scripts were created to perform a series of alignment and reference adjustments using NovoAlign v3.01 (NovoCraft: http://novocraft.com), VarScan v2.3.6 (*Koboldt et al., 2012*) and Mapsembler2 v2.1.6 (*Peterlongo & Chikhi, 2012*) to generate a per individual reference for SNP calling without the need for de novo assembly (Fig. 1B). Perl scripts are available in a github repository (https://github.com/chodon/zingiberales). The plastid sequences were processed the same way except extension with Mapsembler2 was omitted, and individual genes were extracted from the whole plastid prior to final mapping. Finally, reads were mapped with

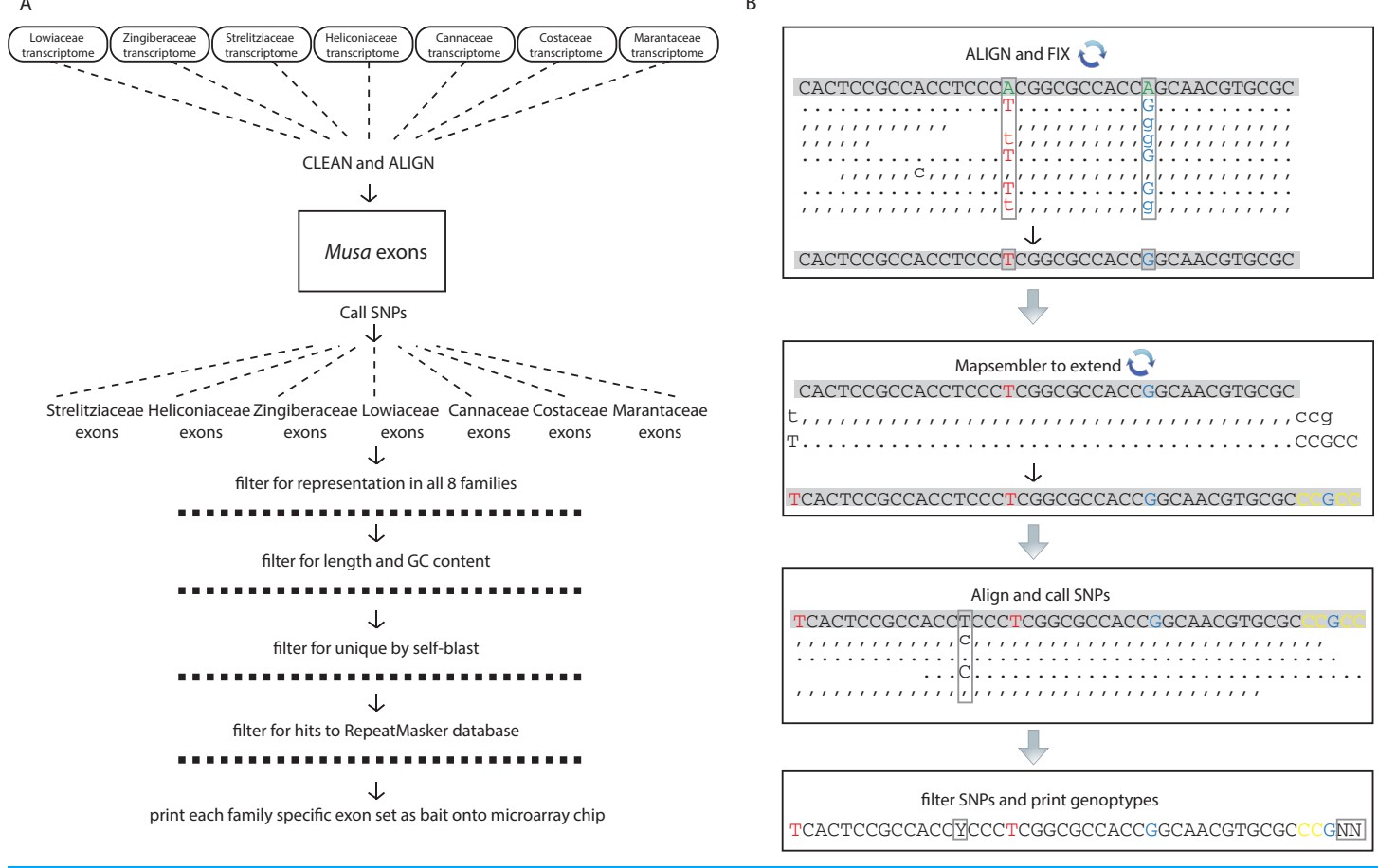

**Figure 1** **Schematic diagrams for the bioinformatic work flow.** (A) Work flow to generate family specific bait sequence from transcriptomes and the annotated exons from *Musa acuminata* and (B) work flow to generate individual sequences for each gene from raw reads independent of de novo assembly. Base changes and SNPs are highlighted and the schematic is represented as in the SAMtools tview format (i.e., reverse reads are represented with commas and lowercase letters). The representation is condensed to show examples of how the reads are transformed but the actual coverage used to call SNPs was at least 20× (see methods).

NovoAlign with –t 90 and PCR duplicates were removed with Picard v1.103 (http://picard.sourceforge.net). SNPs were called following best practices guidelines using the HaplotypeCaller and readBackedPhasing algorithms in GATK v3.1.1 (*McKenna et al., 2010*; *DePristo et al., 2011*; *Van der Auwera et al., 2013*), except quality scores were not recalibrated because the lack of a reference set of known variants. Consensus sequences were created based on SNP calls for regions with greater than 20× coverage (*Nielsen et al., 2011*). SNPs in areas with less than 20× coverage were converted to Ns and regions with less than 5× coverage were discarded. For outgroup taxa, raw reads from transcriptomes generated as part of OneKP were subject to the same pipeline as sequences generated de novo. The raw sequence data from the *Musa balbisiana* genome project (*Davey et al., 2013*) was also subject to the pipeline, but only aligned for the plastid gene set. Raw de novo sequence reads are available on NCBI under bioproject SRP066318 and the final concatenated alignment is accessible on github https://github.com/chodon/zingiberales.

## Alignment

After consensus sequences were made, a second pipeline was made to pass sequences through a series of alignment steps to (1) trim sequences to the *Musa* reference (MAFFT v7.164 (*Katoh et al., 2002*; *Katoh & Standley, 2013*) and mothur v1.34.4 (*Schloss et al., 2009*)), (2) place sequences into coding frame (MACSE v1.01b (*Ranwez et al., 2011*)), and (3) align by codon position (prank v140603 (*Löytynoja & Goldman, 2005*)). Plastid gene introns were spliced out by hand in Geneious v5.6.4 (*Kearse et al., 2012*) prior to step 3, above. After alignment, several additional steps were taken to eliminate genes that might contain non-orthologous sequences. Gene trees were generated with RAxML v8.1.17 (*Stamatakis, 2014*) and the single gene trees were assessed to identify those in which the gene of a single individual taxon accounted for greater than 15% of the total tree length (*dos Reis et al., 2012*). Exon sequences from one individual were BLASTed to the nucleotide collection database (BLASTN v2.2.30+, *Altschul et al., 1997*). Exons were removed from further analyses if significant BLAST hits were found to a whole plastid genome, or to ribosomal, transposon, or mitochondrial DNA. Exons were also removed from further analysis if they had unexpectedly high average coverage of greater than 200× or because frameshifts were introduced during codon position assignment or the alignment had too many indels to be reliable (Table S2). We also manually checked all alignments for potential problems (*Rothfels et al., 2015*). Command parameters for all steps can be found in Supplementary Methods.

## Phylogenetic analyses

The nuclear and plastid sequence data were concatenated and analyzed using maximum parsimony (MP), maximum likelihood (ML), and coalescent approaches. For MP, PAUP* v4.0a142 (*Swofford, 2002*) was used to perform a heuristic search with 100 random addition sequence replicates and default parameters (TBR branch swapping with one tree held per replicate). MP support was evaluated with 1000 bootstrap replicates, each with 10 random addition sequence replicates. For ML reconstruction, gene-by-codon position partitions were created for the complete concatenated data set resulting in a total of 1128 initial partition subsets. These initial subsets were then grouped using the relaxed hierarchical clustering algorithm with a 1% search strategy (*Lanfear et al., 2014*) implemented in PartitionFinder v1.1.1 (*Lanfear et al., 2012*). The resulting partitioning scheme generated by PartitionFinder consisted of 112 subsets (see Supplementary Methods). The PartitionFinder scheme was analyzed with RAxML v8.1.24 (*Stamatakis, 2014*) with the GTR+$\Gamma_4$ model of sequence evolution estimated for each partition subset and the topology linked across partitions. ML support was evaluated for the same partitioning scheme with 1000 bootstrap replicates, using the rapid bootstrap algorithm (*Stamatakis, Hoover & Rougemont, 2008*), and using the CAT$_{25}$ approximation instead of $\Gamma_4$, to model site-to-site rate heterogeneity (*Stamatakis, 2006*). The RAxML analysis was performed on the CIPRES web server (*Miller, Pfeiffer & Schwartz, 2010*). For coalescent analysis, first, best ML gene trees and bootstrap gene trees were generated in RAxML v8.1.17 using the GTR+$\Gamma_4$ model. Best gene trees were the best tree of 20 independent searches and support was evaluated with 1000 bootstrap

replicates. ASTRAL-II v4.7.8 (*Mirarab & Warnow, 2015*) was used to generate an optimal tree based on the best ML gene trees and support evaluated with the 1000 bootstrap replicates.

## RESULTS

### Probe design, sequence capture, and alignment

All targeted regions for all individuals were successfully captured, although average coverage varied based on gene region (Fig. 2A), individual, and phylogenetic distance to the reference sequence (Fig. 2B). Members of the Musaceae, in general, captured better than any other family, likely because they are phylogenetically closest to the original genomic reference upon which the probes were designed. Within each family, close relatives of the species or taxon used to design the bait had higher success rates of capture than more distant members of the family. For example, *Siphonochilus kirkii*, had the lowest average coverage and capture efficiency for Zingiberaceae (Figs. 2B and 2C) as predicted by its evolutionary distance from the transcriptome-sequenced taxon *Curcuma longa*. The minimum percent identity of any captured sequence to its bait was 73%, while average distances were between 94–99% identity (Table S1). Of the total sequenced bases, the capture efficiency varied across individuals with the maximum percentage of bases mapping $3.5\times$ higher than the minimum percentage (Fig. 2C). An average of 26% of captured bases mapped to target, which is similar to capture efficiency reported in captures of human mitochondrial DNA (*Maricic, Whitten & Pääbo, 2010*) and transcriptome based capture of chipmunk DNA (*Bi et al., 2013*). Despite the attempt to capture nuclear and plastid targets evenly, sequencing was highly biased towards plastid targets (Fig. 2C). There was some variability between individuals that was independent of phylogenetic distance, likely due to the standard variation in the success of DNA library preparation, which results from differences in DNA quality, genome size, and difficulties of accurately quantifying DNA for pooling in equimolar quantities. Any differences in DNA concentration were likely amplified in the post-hybridization PCR enrichment step.

Of the 494 nuclear probe exons, 124 were removed from further analyses based on coverage, BLAST results, skewed tree length, or alignment anomalies (Table S2). These 124 exons were from 110 genes. Twenty exons from 14 genes failed a test for coverage outliers because they had greater than $200\times$ coverage, which is outside of the 99.99% confidence interval (Fig. S1). It is possible that these regions were either incorrectly annotated as nuclear regions in the *Musa* draft genome, or were transferred to the nuclear genome from more high copy genomes, especially considering that 15 of these exons were annotated as having an "unknown chromosomal location" in the *Musa* draft genome (Fig. 2A, Table S2). A total of 37 exons from 34 genes were removed from the nuclear dataset and 13 genes from the plastid dataset due to skewed tree length. Four nuclear exons from two genes were removed because of introduced frameshifts and *ycf*1 from the plastid was eliminated because of insertions and deletions in the alignment apparent after manual inspection. Finally, 63 additional exons from 61 genes were removed because of a top BLAST hit to a whole plastid genome, mitochondrial, transposon or ribosomal DNA.

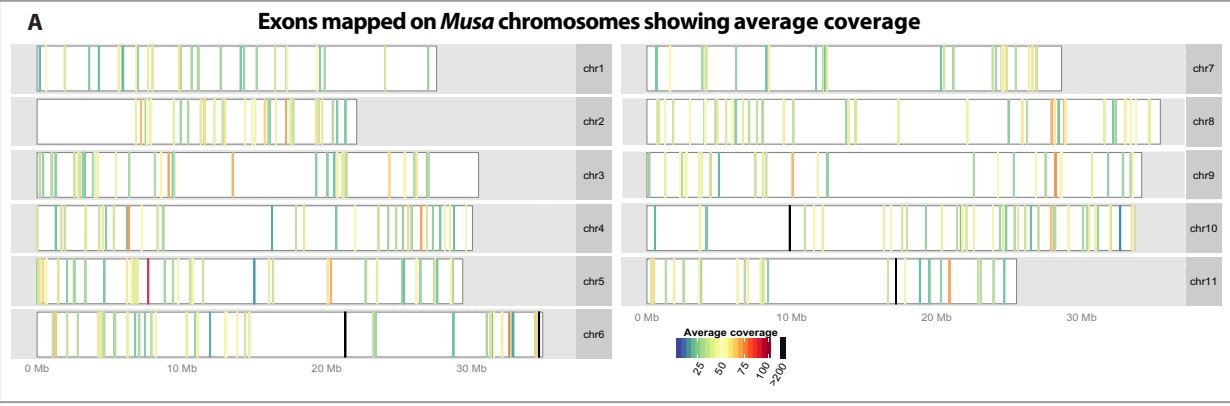

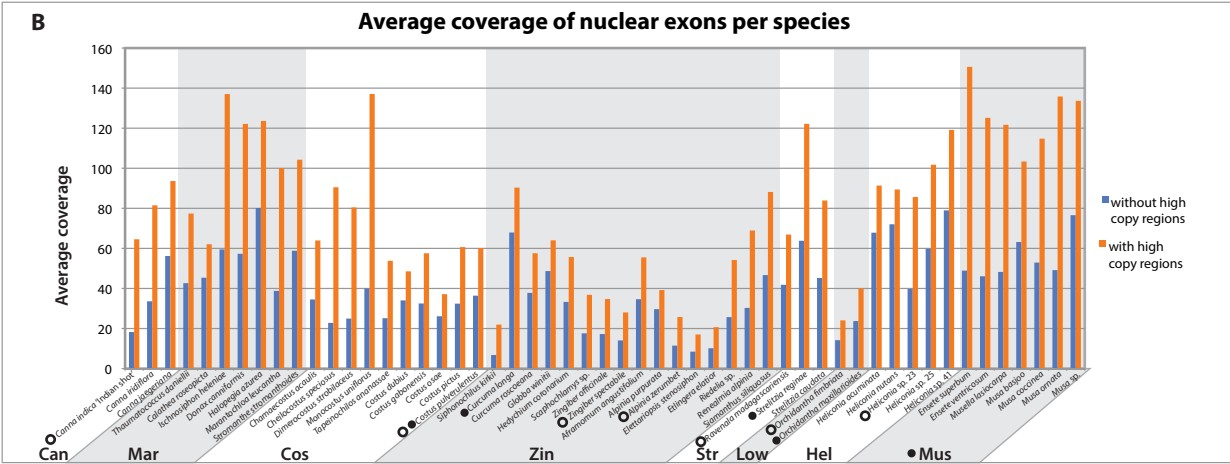

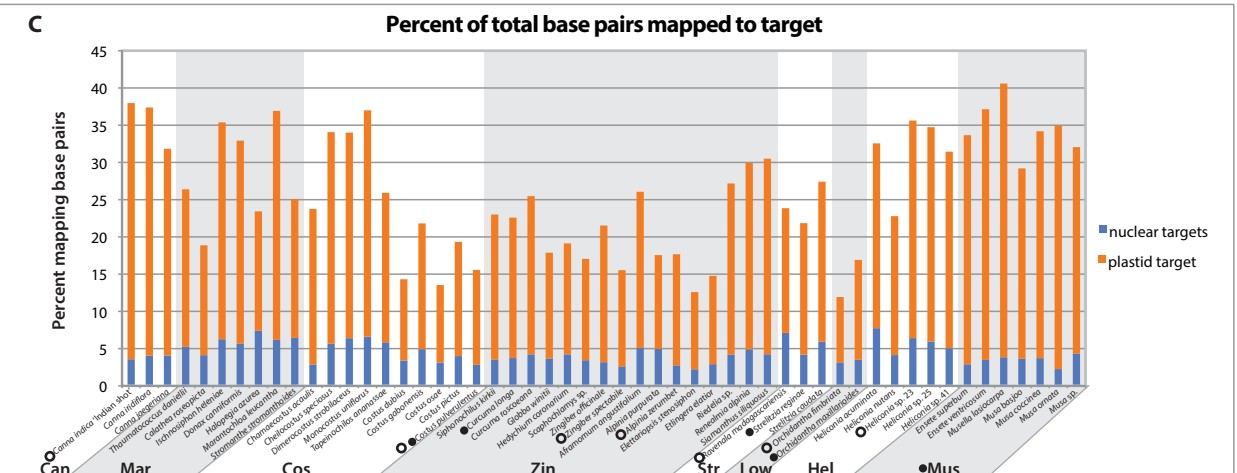

**Figure 2 Capture efficiency across individuals and exons.** (A) Average coverage-depth across all individuals as represented by colored bars placed according to location on *Musa acuminata* chromosomes. Exons with greater than 200× coverage are represented with black bars. Bars representing exons with less than 200× coverage are colored according to average coverage in the scale shown. Exons with unknown chromosomal location are not shown but information can be found in Table S2. (B) Average coverage over all exons for each individual after removing PCR duplicates and with strict alignment parameters that were used for SNP calling. Average coverage was calculated before and after removing the high coverage exons indicated in 2A. (C) Per individual, the percent of the total sequenced base pairs passing Illumina quality filters that mapped to target regions prior to PCR duplicate removal. Percent of base pairs mapping to chloroplast plastid and nuclear regions are indicated in orange and blue, respectively. Species are grouped by family (Can=Cannaceae, Mar=Marantaceae, Cos=Costaceae, Zin=Zingiberaceae, Str=Strelitziaceae, Low=Lowiaceae, Hel=Heliconiaceae, Mus=Musaceae) and species upon which baits were generated are indicated with a filled circle (nuclear bait) or open circle (plastid bait).

Of these 63 exons, the 27 ribosomal and 21 mitochondrial exons could likely be included in further analyses or within family specific analyses in future work after analyzing secondary structure and genomic location.

The final dataset of 308 nuclear genes had a total aligned length of 81,546 bp with 24,379 (29.9%) parsimony informative sites and an average coverage of 40 ± 13× (mean ± s.d.). The 68 gene plastid dataset had a total aligned length of 56,202 bp with 8,336 (14.8%) parsimony informative sites and an average coverage of 377 ± 589× (Table S2).

### Phylogenetic analyses

The recovered topology (Fig. 3) places Musaceae as sister to all other families with 100% parsimony bootstrap support (pb) and maximum likelihood bootstrap support (mlb). The ginger families (Cannaceae, Costaceae, Zingiberaceae and Marantaceae) are well supported (100 pb/mlb) as monophyletic. The MP and ML trees are largely congruent and support values are generally high from shallow to deep phylogenetic relationships. The optimal coalescent tree does not conflict with the ML tree but there is low support for several nodes, notably the placement of Musaceae sister to the rest of the order (Fig. 3).

## DISCUSSION

This method functions to capture hundreds of loci across deep divergence, with successful capture across individual species that are divergent from the genomic data for which the baits were generated. Using several different taxa as bait and filtering genes for those found in all families ameliorated the problem of decreased capture efficiency as phylogenetic distance from probes increases. Furthermore, this protocol can be customized to any plant group and can often be generated with publically available data generated from previous studies. Despite deep phylogenetic divergence, the array-based capture was effective, enabling the avoidance of high efficiency, but costly, in-solution capture protocols. Although the method is limited by the necessity to find orthologs across transcriptomes of varying quality, generated in different labs, and under different conditions, the probe generation and filtration protocol successfully found hundreds of orthologous loci, which offered significant signal at the evolutionary depth of this study. The number of orthologous loci that are expected to be necessary to provide sufficient power to resolve questions asked should be considered when tailoring this pipeline for other systems. Future work will focus on limiting mistaken high copy and excessive plastid capture as well as minimizing the introduction of PCR duplicates.

Family relationships within Zingiberales have been studied since the mid-1950s (Tomlinson, 1956; Tomlinson, 1962). Based on morphological, anatomical, and developmental data a monophyletic 'ginger' clade (Zingiberaceae, Costaceae, Cannaceae and Marantaceae) has long been established (Dahlgren & Rasmussen, 1983; Kirchoff, 1988). However, there are no reliable estimates for the relationships among the other four families (i.e., the 'banana' lineages: Musaceae, Heliconiaceae, Lowiaceae, and Strelitziaceae) and the ginger clade despite several phylogenetic studies from combined

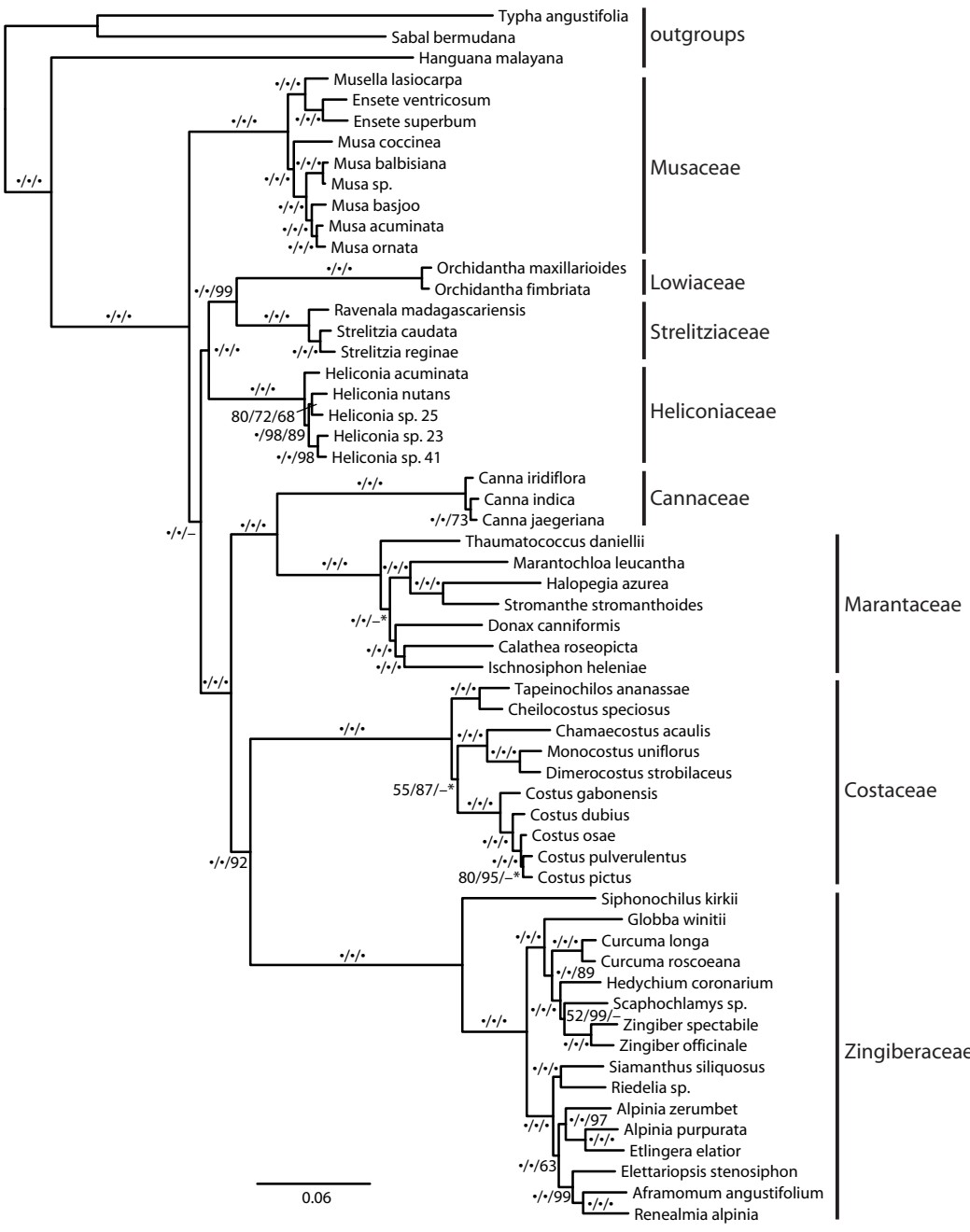

**Figure 3 Phylogenetic tree of Zingiberales based on a partitioned ML of concatenated plastid and nuclear sequence.** Bootstrap support values adjacent to branches as MP/ML/ASTRAL. A dot indicates 100% bootstrap support, a dash indicates less than 50% support and an asterisk indicates conflicting topology in the ASTRAL optimal tree. Scale is in expected substitutions per site.

genomic compartments and morphological data (*Kress, 1990*; *Kress et al., 2001*; *Johansen, 2005*). Even studies using plastome scale datasets failed to produce a well resolved phylogeny near the root of the Zingiberales (*Barrett et al., 2014*). Here, we show that a targeted exon capture generates phylogenomic scale data that can fruitfully address this problem and may be adapted for resolving ancient radiation in other plant groups.

Our main finding suggests that Musaceae is the sister group to the remaining families of Zingiberales and that many other deep relationships within Zingiberales are well supported (Fig. 3). Recent studies of gene family evolution and gene duplication (*Bartlett & Specht, 2010*; *Yockteng et al., 2013*; *Almeida, Yockteng & Specht, 2015*) further support this placement of Musaceae. Relationships within individual Zingiberales families are also well supported in the ML and MP analyses (Fig. 3). The coalescent analysis using ASTRAL-II did not show support for some relationships, but the validity of applying these approaches remains unclear (*Gatesy & Springer, 2013*; *Gatesy & Springer, 2014*; *Mirarab et al., 2014*). Importantly, the relationships found here are not in conflict with existing well supported hypotheses for generic-level relationships (*Kress, Prince & Williams, 2002*; *Johansen, 2005*; *Prince & Kress, 2006*; *Specht, 2006*; *Kress et al., 2007*; *Prince, 2010*; *Li et al., 2010*; *Cron et al., 2012*), indicating that our method is identifying orthologs and that the data produced should be useful at finer phylogenetic scales as well a deep ones.

This pilot study is a first attempt at harnessing phylogenomic data from both the nuclear and plastid genomes to address the global phylogeny of Zingiberales. We have planned substantially increased taxon sampling for both ingroups and out groups and work is ongoing to incorporate morphological data from living and fossil representatives into a phylogenetic reconstruction pipeline to co-estimate fossil placement and lineage divergence times. This will permit us to make full use of information recorded in both the fossil record and genetic data to understand morphological evolution of floral and vegetative traits across the Zingiberales, and estimate ages of diversification for the major lineages, testing the hypothesis of an ancient and rapid radiation at the base of the order.

## ACKNOWLEDGEMENTS

We thank Lydia Smith at the Evolutionary Genetics Lab and Ke Bi at the Computational Genomics Resource Lab at UC Berkeley for invaluable help and discussion with lab and bioinformatics work; Igor Antoshechkin at CalTech for help with the QSonica; the Huntington and UC Botanical Gardens for collections and hosting CS; Jerry Davis for collections and DNA; OneKP and MonAToL, especially Jim Leebens-Mack and Dennis Wm. Stevenson, for early access to raw transcriptome reads used for baits.

### Funding

This research was supported by NSF DEB 1257701 Collaborative Research Award to CDS and SYS, and NSF DEB 0816661 Research Opportunity Award Supplement to CDS and CFB. This work used the Vincent J. Coates Genomics Sequencing Laboratory at UC Berkeley, supported by NIH S10 Instrumentation Grants S10RR029668 and S10RR027303. The funders had no role in study design, data collection and analysis, decision to publish, or preparation of the manuscript.

## Grant Disclosures

The following grant information was disclosed by the authors:
CDS and SYS: NSF DEB 1257701.
CDS and CFB: NSF DEB 0816661.
NIH S10: S10RR029668 and S10RR027303.

## Competing Interests

The authors declare that they have no competing interests.

## Author Contributions

- Chodon Sass conceived and designed the experiments, performed the experiments, analyzed the data, contributed reagents/materials/analysis tools, wrote the paper, prepared figures and/or tables, reviewed drafts of the paper.
- William J.D. Iles performed the experiments, analyzed the data, wrote the paper, prepared figures and/or tables, reviewed drafts of the paper.
- Craig F. Barrett conceived and designed the experiments, performed the experiments, reviewed drafts of the paper.
- Selena Y. Smith conceived and designed the experiments, reviewed drafts of the paper.
- Chelsea D. Specht conceived and designed the experiments, contributed reagents/materials/analysis tools, wrote the paper, reviewed drafts of the paper.

## DNA Deposition

The following information was supplied regarding the deposition of DNA sequences:
NCBI bioproject SRP066318

## Data Deposition

The following information was supplied regarding data availability: https://github.com/chodon/zingiberales and NCBI bioproject SRP066318

## Supplemental Information

Supplemental information for this article can be found online at http://dx.doi.org/10.7717/peerj.1584#supplemental-information.

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
