# Peer review of "Revisiting the Zingiberales: using multiplexed exon capture to resolve ancient and recent phylogenetic splits in a charismatic plant lineage"

_PeerJ, doi:10.7717/peerj.1584_

## Round 0.1 · original submission · Minor Revisions

Both reviewers agree that the manuscript is of excellent quality, and of relevance to both plant systematics, and more generally to methods development. However, Reviewer 2 suggests a few other computational approaches that may help with the analysis. I look forward too seeing the revised manuscript.

·

Basic reporting

Please refer to "General Comments for the Author"

Experimental design

Please refer to "General Comments for the Author"

Validity of the findings

Please refer to "General Comments for the Author"

Additional comments

De novo exon captures have been widely used in animals particularly in vertebrates to address a variety of phylogenetic and population genetic questions. However the same approach has not been as popular in evolutionary applications for plants. Sass et al. used array-based exon capture and HTS to enrich and sequence several hundreds of exonic markers from highly divergent families of Zingiberales to answer a phylogenetic question. Overall this is a solid piece of work. The probe design, wet-lab protocols and bioinformatics are quite rigorous. The merit of this work is obvious: the authors used multiple transcriptome references to extract exonic markers in order to maximize the capture efficiency. This is especially necessary when samples from distinct taxa are pooled and captured on the same chip. But there is a trade-off for using this approach: if there are too many references included in the probe design, we may not end up with enough orthologs shared across all transcriptomes. However, this might not be an issue for resolving high-level phylogenetics since the signals may get saturated after a few hundreds of markers are used. I suggest the authors address this potential issue in the discussion.

Some other comments are listed below:

Line 129: Do authors have data showing the pairwise divergence of these loci between selected taxon? Are these markers kept because they were relatively conserved across all taxa?
Line 147: Singhal & Moritz, 2012 - > Singhal 2012
Line 152: The link https://github.com/chodon/zingiberales does not work!
Line 156: Was GATK readBackedPhasing used to phase the sequencings?
Another question: is there a particular reason why the authors chose not to use SAMTools for SNP calling which may be more suitable for non-model organisms?
Lines 160-161: I am a little confused by this bioinformatics step: Why were SNPs with coverage than 20X converted to “N”s but why were regions (without SNPs) with less than 5x removed? I guess my question is that why two different thresholds were applied depending on if a site is a variable or not.
Line 236-237: “twenty exons from 14 genes had greater than 200× average coverage suggesting that these regions are part of highly repetitive areas” – Exon capture data are highly heterogeneous in nature therefore loci with high coverage are not necessarily repetitive. I agree that it is generally a good idea to purge loci with extreme coverage. 200x sounds quite arbitrary though. Is this based on some kind of statistics? For instance, does 200x fall in top 5% of the coverage distribution?
Line 250-252: what is the average depth and variance of the loci kept for phylogenetic analyses?

·

Basic reporting

This is an excellent piece of work that offers advances in both methods for analyzing plant exon capture data and understanding of relationships within the Zingerberales. The published manuscript will be of broad interest to the plant systematics community and those working on economically, culturally and ecological important species within the family. I have just a couple of suggestions that are intended to help readers better interpret the very clearly presented results of this study.

Most importantly, as the authors acknowledge, the rapid series of speciation events that have been so challenging for systematists to resolve despite decades of work may confound supermatrix analyses in the face of incomplete lineage sorting (ILS). Whereas uncertainty in gene tree estimation is indeed problematic for most coalescence-based species tree estimation approaches, the high degree of confidence implied in the bootstrap support values for the supermatix tree may be misleading. I encourage the authors to use one or more coalescence-based species tree estimation algorithms (ASTRAL, MP-EST or SVDquartets) and offer an interpretation for any conflict between the supermatrix and coalescence-based tree estimates, or possible lack of resolution in a coalescence-based tree inference. If there is very little resolution in the gene trees, SVDquartets does not rely on gene tree estimates. SVDquartets does assume independence among SNPs but the approach has been shown to be robust in some cases where there is a lack of independence among SNPs within and exon or linked exons.

The other minor point is that divergence time estimates for the origin and diversification of the Zingiberales vary among studies. For example Magallon et al. (2015, New Phytologist) recently reported younger age estimates, although their 95% credibility interval encompasses the age estimates of Kress and Specht. Nonetheless, readers should be provided with some understanding of the variation in divergence time estimates.

Experimental design

Aside from the analytical issues relating to ILS, the experimental design is well reasoned and well described.

Validity of the findings

See comments above.

Additional comments

I look forward to publication of this excellent work!

---

## Round 0.2 · accepted · Accept

The manuscript was already in good shape upon arrival, both the reviewer and I feel that the recent changes made it stronger. I recommend that it be published as it, and ask the authors to release all the data and analysis scripts.

·

Basic reporting

This is a re-submitted MS. The authors have very well addressed my questions and comments made to the previous version and I am happy about their effort in improving the MS.

Experimental design

No further comments

Validity of the findings

no further comments